# Alpha-Synuclein in Peripheral Tissues as a Possible Marker for Neurological Diseases and Other Medical Conditions

**DOI:** 10.3390/biom13081263

**Published:** 2023-08-18

**Authors:** Félix Javier Jiménez-Jiménez, Hortensia Alonso-Navarro, Elena García-Martín, Diego Santos-García, Iván Martínez-Valbuena, José A. G. Agúndez

**Affiliations:** 1Section of Neurology, Hospital Universitario del Sureste, Arganda del Rey, 28500 Madrid, Spain; 2Institute of Molecular Pathology Biomarkers, Universidad de Extremadura, 10071 Cáceres, Spain; elenag@unex.es (E.G.-M.); jagundez@unex.es (J.A.G.A.); 3Department of Neurology, CHUAC—Complejo Hospitalario Universitario de A Coruña, 15006 A Coruña, Spain; diegosangar@yahoo.es; 4Tanz Centre for Research in Neurodegenerative Diseases, University of Toronto, Toronto, ON M5T 2S8, Canada; ivan.martinez@utoronto.ca

**Keywords:** alpha-synuclein, peripheral tissues, parkinson’s disease, alpha-synucleinopathies, biological markers

## Abstract

The possible usefulness of alpha-synuclein (aSyn) determinations in peripheral tissues (blood cells, salivary gland biopsies, olfactory mucosa, digestive tract, skin) and in biological fluids, except for cerebrospinal fluid (serum, plasma, saliva, feces, urine), as a marker of several diseases, has been the subject of numerous publications. This narrative review summarizes data from studies trying to determine the role of total, oligomeric, and phosphorylated aSyn determinations as a marker of various diseases, especially PD and other alpha-synucleinopathies. In summary, the results of studies addressing the determinations of aSyn in its different forms in peripheral tissues (especially in platelets, skin, and digestive tract, but also salivary glands and olfactory mucosa), in combination with other potential biomarkers, could be a useful tool to discriminate PD from controls and from other causes of parkinsonisms, including synucleinopathies.

## 1. Introduction

Polymeropoulos et al. [1], in 1997, described mutations in the gene encoding the presynaptic protein alpha-synuclein (aSyn) as the first mutations related to autosomal dominant Parkinson’s disease (PD). In the same year, Spillantini et al. [2] described the presence of aSyn aggregates in Lewy bodies (the pathologic hallmark of PD) [2]. Subsequently, aSyn aggregates were also described in other diseases termed “synucleinopathies”, such as PD with dementia (PDD), dementia with Lewy bodies (LBD), and multiple system atrophy (MSA; in this entity, the aggregates were present in the form of glial cytoplasmatic inclusions) [3]. Since then, many investigators have made important efforts to study the possible role of determinations of this protein in biological fluids and other tissues as a potential biomarker of PD and other synucleinopathies [4].

Although the first attempt to find aSyn in the cerebrospinal fluid (CSF) of PD patients and healthy controls was unsuccessful [5], further studies described its presence in the CSF [6] and in other extracellular biological fluids, including plasma [7]. A recent meta-analysis of 22 eligible studies addressing CSF aSyn levels in a total of 1855 patients with synucleinopathies and 1378 control subjects clearly showed high sensitivity and specificity to this value to differentiate synucleinopathies from controls, although the specific results for MSA were not robust enough [8].

The development of early biomarkers of PD and other synucleinopathies is very important to try to establish possible preventive treatments for these diseases. Regarding this topic, the possible value of aSyn determinations in other peripheral tissues in PD, in other synucleinopathies, and other neurological diseases has been the subject (especially in PD) of numerous studies. In this narrative review, we summarize data from studies on aSyn concentrations in different peripheral tissues, in PD, in other parkinsonian syndromes and in other neurological diseases, as well as the effect of age/aging on these. For this purpose, we conducted a literature review using the PubMed database from 1966 to 21 July 2023, by crossing the terms “alpha synuclein” and “levels” and “concentrations” with “serum” (266 items), “plasma” (377 items), blood cells (244 items), erythrocytes (42 items), lymphocytes (54 items), monocytes (26), leukocytes (76 items), platelets (35 items), “stool” (13 items), “sweat” (2 items), and “urine” (13 items); and the term “alpha-synuclein” with “salivary glands biopsy” (40 items), “saliva” (39 items), “olfactory mucosa biopsy” (27 items), “gastrointestinal tract biopsy” (227 items), “colon biopsy” (126 items), “stomach biopsy” (36 items), “duodenum biopsy” (11 items), “small intestine biopsy” (27 items), “skin biopsy” (153 items). The whole search retrieved a total of 957 references that were manually examined by the authors to select publications strictly related to this issue.

## 2. Studies Addressing Serum/Plasma aSyn Levels

### 2.1. Parkinson’s Disease

The results of studies addressing serum and/or plasma aSyn levels in PD patients compared with controls are summarized in Table 1 [9,10,11,12,13,14,15,16,17,18,19,20,21,22,23,24,25,26,27,28,29,30,31,32,33,34,35,36,37,38,39,40,41,42,43,44,45,46,47,48,49,50,51,52,53,54,55,56,57,58,59,60,61,62,63,64,65,66,67]. Thirty-one studies have shown increased serum and/or plasma total aSyn levels [10,16,17,27,30,31,33,38,41,42,44,46,47,48,49,50,51,53,54,57,61,63,65], serum and/or plasma oligomeric aSyn levels [45,52,59], or serum and/or plasma aSyn antibodies levels in PD patients compared to healthy controls [13,15,17,26,36]. Twenty-two studies showed similar results for PD patients and controls in serum and/or plasma aSyn [11,12,18,21,22,23,29,32,37,39,40,56,60,66] or aSyn antibodies levels [9,14,24,25,28,35,43,62], and another six studies showed decreased serum/plasma aSyn levels [19,55,58] or aSyn antibodies levels in PD patients [20,34,64]. Interestingly, three studies have shown increased levels of aSyn in serum or plasma exosomes (i.e., small extracellular vesicles that carry many proteins, lipids, or miRNA) from PD patients compared with controls [27,60,61], with total serum aSyn levels being similar in PD patients and controls [60]. A recent meta-analysis including 32 eligible studies addressing total, oligomeric, or phosphorylated aSyn in the plasma and/or serum of PD patients and healthy controls, involving 2683 PD patients and 1838 controls, showed increased total plasma/serum aSyn levels in PD patients, and a lack of significant differences in oligomeric aSyn between the two study groups [68]. Five studies addressing serum/plasma phosphorylated aSyn showed an increase in this value in PD patients [22,44,53,65,66].

Several studies described an association between serum/plasma total aSyn and cognitive impairment, hallucinations and sleep disorders [21], a negative correlation between aSyn levels and PD severity [54], lower serum aSyn levels in patients with advanced PD [11,13], and higher serum plasma total aSyn in PD patients with predominant postural instability and gait difficulties compared with those with tremor-dominant PD [33,65], and in those patients with an infectious bacterial [30] and viral [30,31] burden. Serum/plasma aSyn levels were found to be correlated with increased serum sirtuin 2 [42], Rab35 [47], nod-like receptor protein 3 [48] levels, and decreased serum mortalin levels [38].

### 2.2. Other Parkinsonian Syndromes

The results of studies measuring serum/plasma aSyn levels in patients with other parkinsonian syndromes are summarized in Table 2.

Most of the studies on patients diagnosed with MSA compared with controls showed non-significant differences in serum/plasma aSyn levels between the two groups [43,47,61,62], while others showed a significant increase [10,42] or significant decrease [34] in this value in MSA patients. Four studies showed a lack of significant differences in aSyn levels between PSP patients and healthy controls [47,60,61,62].

Two studies showed similar serum/plasma aSyn levels in patients with DLB and controls [60,61], while another three showed increased [53,69,70,71] and decreased [70] aSyn in DLB patients. Concerning patients with PDD, three studies have shown increased serum/plasma aSyn [53,57,69], one study showed decreased aSyn [72], and another showed a lack of differences when compared with healthy controls [24]. Kronimus et al. [73] described lower serum aSyn levels in patients with PDD compared with non-demented PD patients. aSyn levels in serum [62] and in serum exosomes [61] were found to be similar in patients with CBD and controls.

Patients with coexistent REM sleep behaviour disorder and PD showed significantly higher serum aSyn levels than patients with PD without RBD in a single study [74].

### 2.3. Other Neurological and Neuropsychiatric Diseases

Serum aSyn levels are increased in patients with Huntington’s disease (34 symptomatic and 4 premanifest) compared with controls (*n* = 36) and were not related to the treatment received, age, disease duration, and severity of the disease, suggesting a possible role of the interaction between mutant huntingtin and aSyn aggregates in this disease [75].

Several studies have shown similar serum aSyn levels [11,62,70] or antibodies against aSyn levels [20,69,72] in patients with Alzheimer’s disease (AD) [11,20,62,69,70,72] and similar serum aSyn antibodies levels in patients with vascular dementia [69,72] than those of healthy controls. In contrast, other studies described increased plasma aSyn levels in patients with AD compared with healthy controls [57,76], and another two studies showed increased autoantibodies against aSyn in AD patients [36,71], although fewer than those found in PD patients [71]. Patients with frontotemporal dementia also showed similar aSyn levels when compared with controls [62,69].

Rong et al. [77], in a study involving 67 epileptic patients (40 with intractable, 13 with newly diagnosed, and 14 with non-intractable epilepsy) and 22 controls (diagnosed with neurosis), described increased serum (and CSF) aSyn levels in patients with intractable epilepsy when compared to controls and to the other two groups of epileptic patients. Choi et al. [78] also found increased serum and exosome aSyn levels in 115 children with epilepsy (correlated with disease severity) and in 10 children with acquired demyelinating disorders of the central nervous system compared with 146 healthy matched controls.

A study involving 60 patients diagnosed with multiple sclerosis (83% of them with the relapsing–remitting type) and 60 healthy matched controls showed a significant decrease in serum levels of both total and oligomeric aSyn, and an increase in serum oligomeric aSyn/total aSyn ratio in multiple sclerosis patients [79]. Based on data from previous experimental studies, the authors suggested a possible role of aSyn in the development of neuroinflammation and the diffuse neuronal and synaptic loss [79].

Three studies have shown a significant decrease in serum [80,81] or plasma [82] aSyn levels in children diagnosed with autism spectrum disorders compared with controls, one of them also found decreased serum aSyn levels and *alpha-synuclein* (*SNCA*) gene expression in mothers of patients with autism [82], and other increased plasma beta-synuclein (bSyn) levels in patients with autism [82]. In contrast, another study with a lower sample size reported increased serum levels of autoantibodies against aSyn in autistic patients and their mothers compared with controls and their mothers, respectively [83]. Finally, another two studies from the same group showed increased plasma aSyn [84,85] and decreased plasma gamma-synuclein (gSyn) [84] in patients with autism compared with controls, and a relationship between plasma aSyn and gSyn and disease severity [84].

Serum aSyn levels were reported to be similar in children with attention deficit hyperactivity disorders (*n* = 25) compared with healthy children (*n* = 25) [86].

Maetzler et al. [87] described similar serum IgG autoantibody titers against aSyn in 214 individuals with depression compared with 419 controls, while they found decreased titers of serum IgG autoantibodies against amyloid beta1-42 in depressed patients. In contrast, Ishiguro et al. [88] described increased serum aSyn levels in patients with major depressive disorder (*n* = 103) compared with healthy controls (*n* = 132).

Demirel et al. [89] described decreased serum aSyn levels by approximately three-fold in 44 patients diagnosed with schizophrenia compared to 40 healthy controls. Similarly, Göverti et al. [90] found decreased serum aSyn levels, not only in 62 schizophrenic patients but also in 56 asymptomatic siblings, compared with 56 controls.

### 2.4. Other Medical Conditions

Rodríguez-Araujo et al. [91], in a study involving 1.152 patients, showed an inverse correlation between serum aSyn levels and indicators for insulin resistance, including body mass index, homeostatic model assessment for insulin resistance (HOMA-IR) and immunoreactive insulin (IRI). In addition, they described, in the aSyn knock-out mice model, alterations in glucose and insulin responses during diet-induced insulin resistance, and the development of severe insulin resistance after feeding with a high-fat diet.

In patients with Gaucher’s disease (related to glucocerebrosidase (GBA) deficiency), plasma oligomeric aSyn levels were related to leucocyte GBA activity [92], although plasma and exosomal aSyn levels were similar in patients with PD associated with *GBA* gene mutations, PD without GBA mutations and controls [93].

In 104 patients undergoing elective laparoscopic surgery, serum aSyn levels were not associated with the development of postoperative emergence delirium [94].

Bönsch et al. [95] described the increased serum expression of aSyn protein by approximately three-fold in 49 male alcoholic subjects compared with 50 nondrinking male healthy controls, which was correlated with alcohol craving scores. Similarly, increased serum aSyn protein levels were reported in recently abstinent cocaine-dependent patients (*n* = 38) compared with controls (*n* = 14), with aSyn levels correlating with the intensity and frequency of cocaine-craving episodes [96].

Individuals exposed to pesticides (*n* = 50) have shown a 4.55-fold increase in serum reactivity to aSyn autoantibodies compared to non-exposed subjects (*n* = 25) in a single study [97].

Based on the fact that aSyn was described as a native antiviral factor within neurons and is upregulated in animals with neuroinvasive infections, Blanco-Palmero et al. [98] analyzed serum and CSF aSyn levels in 7 patients suffering from COVID-19 with neurological involvement (myoclonus, parkinsonism, and encephalopathy), 30 with COVID-19 without neurological involvement and 8 healthy controls, finding similar values in the three groups.

### 2.5. Aging

Koehler et al. [99] showed a significant effect of aging in plasma aSyn levels by comparing a group of 40-year-olds with another group composed of 40 young healthy males, with the results for the elder group being approximately three-fold higher than that of the younger subjects. Another study showed similar serum aSyn reactive antibodies in children (*n* = 37) and adults (*n* = 37), although female children had a trend towards an increase in this value compared with female adults [100].

## 3. Studies Addressing aSyn Levels in Blood Cells

### 3.1. Erythrocytes

Because it has been shown that approximately 99% of the aSyn in human blood is present in peripheral blood cells, mainly in erythrocytes, the determination of aSyn levels in these cells is potentially more interesting than the determination in serum/plasma [101].

Studies in PD patients compared with controls have found increased levels of total and oligomeric aSyn [46,102,103,104,105], and 129Ser-phosphorylated aSyn [106] in the membrane of erythrocytes of PD patients. Erythrocyte levels of 129Ser-phosphorylated aSyn were higher in patients with late-onset PD and in PD patients with postural instability-gait disorder [106]. Tian et al. [103] reported significantly higher levels of total and aggregated aSyn in the membrane fraction of erythrocytes from 225 PD patients compared with those of 133 HCs, and significantly higher levels of phosphorylated-Ser129 aSyn in the cytosolic and, to a lesser extent, in the membrane fraction of erythrocytes from PD patients compared with those of HCs. Vicente-Miranda et al. [107] found increased erythrocyte levels of Y125-phosphorylated, Y39-nitrated and glycated aSyn, and decreased sumoylated aSyn in 58 patients with PD compared with 30 HCs. Finally, Papagiannakis et al. [108] described increased dimeric and dimeric/monomeric ratios of aSyn in the erythrocyte membranes from patients with PD carrying glucocerebrosidase mutations and in patients with genetically undetermined PD, but not in PD patients carrying the A53T mutation in the SNCA gene, compared with HCs, while monomeric aSyn levels did not differ significantly between the three groups of PD patients and HCs.

Three studies involving a total of 240 patients diagnosed with *MSA* and 382 HCs found increased total aSyn [102,109,110], oligomeric aSyn [102], and oligomeric aSyn/total red blood cells [103] in patients with MSA.

Baldacci et al. [111] reported lower concentrations of aSyn and its heterocomplexes (aSyn/Aβ and aSyn/tau) in the membrane of erythrocytes of 38 patients diagnosed with AD compared to 38 HC. In the same line, Daniele et al. [112] found significantly lower aSyn and aSyn/tau heterodimers in 51 patients with AD and 27 with LBD compared with 60 HCs. In contrast, Graham et al. [113] found increased concentrations of aSyn in 6 patients with DLB compared with 60 HCs, but normal values in patients with AD and vascular dementia.

One study described that erythrocyte monomeric aSyn levels were similar in 27 patients with Gaucher’s disease and in 32 HCs, although the ratio of dimeric to monomeric aSyn was increased in these patients [114]. Another study described increased erythrocyte aSyn levels in both patients with Gaucher’s disease and asymptomatic carriers [115].

### 3.2. Leukocytes

Brighina et al. described similar aSyn levels in lymphomonocyte cells from patients with sporadic *PD*, and controls [116,117], but lower aSyn levels in a small series of patients carrying mutations in the *LRRK2* gene [117]. Similarly, Wijeyekoon et al. [55] showed a similar monocyte aSyn uptake and aSyn secretion in PD patients and HCs. In contrast, Fan et al. [50] showed an increased mRNA expression for aSyn in peripheral mononuclear cells, and Emelyanov et al. [118] showed a significant increase in aSyn levels in peripheral blood CD45+ cells in 458 PD patients compared with 353 HCs, related to rs356219 and rs356168 variants in the *SNCA* gene.

Patients with relapsing–remitting multiple sclerosis have shown a similar percentage of aSyn-positive lymphocytes and monocytes than HCs [119], with results that were not statistically different between the study groups. Patients with alcohol-dependence syndrome have shown increased mRNA expression for aSyn in the peripheral lymphocytes [120], and patients with juvenile neuronal ceroid lipofuscinosis (Batten disease) have shown enhanced levels of aSyn oligomers in lymphoblast cells [121].

### 3.3. Platelets

Li et al. [122] reported similar platelet aSyn and gSyn levels in PD patients and controls. However, Michell et al. [123] described a high variability in platelet aSyn levels that was not correlated with disease presence or severity. Mukaetova-Ladinska [124] described similar platelet aSyn levels in 25 patients with AD and 26 HCs.

## 4. Studies Addressing aSyn Levels in Salivary Glands and Saliva

### 4.1. Parkinson’s Disease

The first description of the presence of Lewy pathology in salivary glands was carried out by Del Tredici et al. [125], who reported the presence of aSyn using immunocytochemistry in the submandibular gland in 9 of 9 cases of PD, and in 2 of 3 patients with incidental Lewy body disease, but in neither of 2 patients with MSA and 19 HCs. Cersósimo et al. [126] described immunoreactivity to aSyn in labial salivary glands from 3 patients with PD.

A further study in autopsy-proven cases involving 28 PD, 5 incidental Lewy body disease, 5 progressive supranuclear palsy (3 with concurrent PD), 3 corticobasal degeneration, 2 multiple system atrophy, 22 AD with Lewy bodies, 16 AD without Lewy bodies, and 50 normal elderly, found immunoreactivity for aSyn in the submandibular glands of all the 28 patients with PD, the 3 patients with concomitant PD and PSP, and 3 of the patients with AD with Lewy bodies [127]. Chahine et al. [56] found aSyn immunoreactivity in the submandibular glands in 56% of PD patients and 7% of controls, and Vilas et al. [128] found aSyn immunoreactivity in the nerve fibers from submandibular glands in 67% of 12 PD patients and 0% of 26 HCs. Similarly, Shin et al. [129] described immunoreactivity for Ser129-phosphorylated aSyn in 56.2% of 16 PD patients and in none of their 14 controls. The density of immunoreactivity to aSyn in a second biopsy or submandibular gland performed approximately 4 years after the first biopsy increased four-fold according to one study [130]. Manne et al. [131], using a real-time quaking-induced conversion assay (RT-QuIC), showed increased aSyn pathology in the submandibular glands from patients with PD and incidental Lewy body disease compared with HCs. Fernández-Espejo [132] found 3-nitrotyrosine-aSyn immunoreactivity in the submandibular gland from PD patients but not in HCs. As should be expected, patients with PD associated with *parkin* gene mutations (who do not show Lewy pathology) showed a lack of aSyn immunoreactivity in the submandibular glands [133].

The possible value of the studies in minor salivary glands, such as the labial glands, has been questioned by other groups, who found positivity rates of 6.7–18.8% in PD patients [134,135], but 75–77% positivity in submandibular gland biopsy [135,136]. However, other investigators, with sufficient biopsy material, showed a positivity rate of 55–69% in minor salivary glands from PD patients and in 0–38.9% of HCs [137,138,139]. Carletti et al. [140] described a slightly decreased ratio of nerve fibers that were immunoreactive to aSyn in nerve fibers from minor salivary glands in PD patients compared with HCs, but they identified Ser129-phosphorylated aSyn in 5 of 7 PD cases and in none of their HC patients. Similarly, Ma et al. [141] found positive nitrated aSyn in 100% of 8 PD patients and none of 7 HCs.

The description of aSyn immunoreactivity in the salivary glands of PD patients led many investigators to determine aSyn concentrations in the saliva. Five studies found decreased total salivary concentrations in PD patients compared with HCs [142,143,144,145,146], while another six studies found similar values in both study groups [37,56,132,147,148,149]. However, salivary oligomeric aSyn levels [144,145,147,148,149], and oligomeric aSyn/total aSyn ratio [148] were found to increase in PD patients in all the studies addressing this issue. Salivary Ser129-phosphorylated aSyn was similar for PD patients and HCs in one study [148], but was increased in PD patients in another [150], and 3-nitrotyrosine-aSyn did not differ significantly between PD patients and HCs [132]. A meta-analysis of studies reported up to August 2021 described a significant reduction in the mean differences of total salivary aSyn, and a significant increase in the mean difference of oligomeric aSyn and oligomeric/total aSyn ratio in patients with PD compared with HCs [151,152].

Finally, two recent studies using the RTQuIC detection of aSyn in saliva have shown a rate positivity of 76–86% in PD patients and 5.6–22% in HCs [153,154], and one of them suggested a correlation between salivary aSyn and disease severity in de novo PD patients [153].

### 4.2. Other Alpha-Synucleinopathies and Tauopathies

Luan et al. [152] described a positivity rate of RTQuIC for aSyn of 61.1% in the saliva from patients with MSA, which was significantly higher than that found in HCs (5.6%) but did not differ significantly when compared to PD patients (76.0%). Cao et al. [154] reported a significantly lower total aSyn, but similar oligomeric and Ser123-phosphorylated aSyn, in the extracellular vesicles from the saliva of 16 patients with MSA-P compared with 26 PD patients.

Vivacqua et al. [145] found similar salivary total and oligomeric aSyn concentrations in patients with PSP and HCs.

Vilas et al. [128] described aSyn immunoreactivity in the nerve fibers from submandibular glands in 89% of 9 patients with idiopathic REM sleep behaviour disorder (iRBD) and none in 26 controls. The same group described aSyn immunoreactivity in the labial minor salivary glands in 50% of 62 patients with RBD, in 50% of 10 patients with DLB, and 3% of 33 controls [137]. Finally, Mangone et al. [138] described aSyn immunoreactivity in 43.8% of patients with iRBD and 38.9% of their controls.

### 4.3. Other Neurological Diseases

Sabaei et al. [147] described decreased salivary total aSyn concentrations in 24 patients with mild AD compared with 22 HCs, with these concentrations being similar to those found in 24 early PD patients.

A study involving 40 children diagnosed with autism spectrum disorder and 40 HCs showed a significant decrease in both monomeric and oligomeric aSyn concentrations in the saliva in the patient’s group [155]. Finally, salivary oligomeric aSyn levels are increased in symptomatic patients with familial transthyretin amyloidosis associated with mutations in the transthyretin gene compared with asymptomatic carriers with these mutations [156].

## 5. Studies Addressing aSyn Levels in the Olfactory Mucosa

The first study of biopsies from olfactory mucosa did not find immunoreactivity for aSyn in seven PD patients [157]. One post-mortem study involving 79 cases diagnosed with AD, 63 with other neurodegenerative diseases, and 45 neuropathological normal cases found aSyn by immunohistochemistry in 7 cases (2 with DLB, 2 with mixed pathology of AD and LB, 1 PD, 1 AD, and 1 normal) [158]. Other post-mortem study showed immunoreactivity for phosphorylated aSyn in the central or peripheral nervous system from 39 of 105 unselected subjects, and the presence of aSyn deposits in 7 subjects (6 of 8 patients diagnosed neuropathologically with DLB and 1 of 31 with incidental or subclinical LB disease) [159,160].

However, in recent years, several studies using RTQuIC assays in the olfactory mucosa have shown positivity rates of 46.3–69% in patients with PD [161,162,163], 82–90% for MSA-parkinsonism (MSA-P) [161,163], 10.0% for MSA-cerebellar (MSA-C) [163], 81.4 for DLB alone, prodromal or associated to AD (84.4% for DLB alone) [164], 16% for PSP and CBD [161], 44.4% for isolated RBD [162], 7.9% for a mix of patients with non-aSyn related neurodegenerative and non-neurodegenerative disorders [164], and 9.1–10.0% of HCs [163,165]. Another recent study using RTQuIC has shown that positivity in PD patients could be increased from 45% to 84% if the biopsy was obtained from agger nasi instead of the middle turbinate area of the olfactory neuroepithelium [165].

More recently, Schirinzi et al. [166] described increased levels of oligomeric aSyn, but similar levels to total aSyn in the olfactory mucosa from 38 PD patients compared with 31 HCs, with the levels of oligomeric aSyn being correlated with prokineticin-2 protein levels. Kuzkina et al. [167], using seed amplification assays, showed the detection of aggregates of aSyn in the nasal brushing of 48.1% of 27 PD patients, 67% of 18 iRBD patients, and in 1 of 3 MSA and 0 of 3 PSP patients.

## 6. Studies Addressing aSyn Levels in the Gastrointestinal Tract

### 6.1. Parkinson’s Disease

Lebouvier et al. [168] described, for the first time, the presence of immunoreactivity to phosphorylated aSyn in the nerve fibers of the colonic submucosa from 4 out of 5 patients with PD and in none of 5 healthy controls and 3 patients with chronic constipation. Later on, the same group described the presence of immunoreactivity for phosphorylated aSyn in the gastric and duodenal submucosa in one PD patient [169]. Several studies reported in the 2010s found percentages of positive immunoreactivity for aSyn in colonic biopsies from PD patients ranging from 55.5 to 100% [56,170,171,172,173,174] and in none of the control subjects [170,171,172], or patients with inflammatory bowel disease [171]. Moreover, the presence of aSyn was described in 3 PD patients 2–5 years before the onset of motor symptoms [175], and in 4 of 6 subjects with prodromic PD (compared with 5 of 7 PD patients and 2 of 17 controls) [176] who had undergone colon biopsy for other reasons. However, some of these studies showed rates of aSyn positivity in 52% of healthy controls as well [174].

Visanji et al. [177], using a paraffin-embedded tissue blot method, showed the presence of aggregated aSyn and Ser129-phosphorylated aSyn in the colonic mucosa from 80% of patients with early PD and in 100% of patients with later PD and in controls, suggesting the lack of specificity of aSyn as a marker for PD. A similar conclusion was obtained by other investigators, who described 99.7% of immunoreactivity for aSyn in the colon from PD patients and 100% in controls [178], and similar (although lower) percentages of immunoreactivity for aSyn in the colon [179,180] and in the stomach [179] from PD patients compared with controls, with one of them using different methods [180]. Beck et al. [181] described immunoreactivity for aSyn in the colon and stomach in 100% of premotor PD patients and 82% of controls, but immunoreactivity for phosphorylated aSyn in the nerve fibers from the muscularis propria was present in 100% of premotor PD patients and only 14% of controls.

Hilton et al. [182] reported positivity rates for aSyn of 13.2% in the colon, 8.6% in the stomach, 13.3% in the small intestine, and 0% in the esophagus and gallbladder from PD patients (in some patients before development of motor symptoms), while none of their controls showed immunoreactivity. Aldecoa et al. [183], using several types of primary antibodies against aSyn, showed higher rates of immunoreactivity in the stomach, small bowel, and colon from PD patients (50–66.7%) than in controls (8.3–25%). Other authors identified higher rates of positive immunoreactivity for phosphorylated aSyn in PD patients compared with controls in the stomach (58.3% vs. 8.3%) but similar rates in the colon (23.8% vs. 23.8%) [184], or non-significant differences between PD patients and controls in the rates of positivity for aSyn in gastric and colonic biopsies (43.5% vs. 36.4%) [185]. The positivity of aSyn immunostaining in gastric or colonic mucosa in PD patients has been associated with the allelic variant rs11931074G in the *SNCA* gene, but not with longer *SNCA* Rep1 alleles [186].

Beach et al. [187], in a study involving 5 PD patients and 5 controls using 7 different immunohistochemical methods, described that when sufficient colonic mucosa and Lewy-type alpha-synucleinopathy is present, adequately trained raters could distinguish PD from controls. Emmi et al. [188] found immunoreactivity for aggregated aSyn in the duodenum in 100% of 22 PD patients (18 of them with advanced PD) and 0% of 18 controls.

A meta-analysis on the diagnostic utility of determination of colonic aSyn immunoreactivity including eligible studies from up to June 2018 found rates of sensitivity and specificity for distinguishing PD and controls of 0.568 and 0.819 for aSyn and 0.579 and 0.822 for phosphorylated aSyn, respectively [189], suggesting a possible role of gut aSyn as a marker for PD in combination with other biomarkers for this disease.

Kim et al. [190], described positive immunostaining for aSyn and phosphorylated aSyn in 10 of 12 PD patients who underwent colectomy for colorectal cancer, with the positivity being related to the presence of 2–7 of 16 premotor markers for PD.

### 6.2. Other Synucleinopathies

One study showed a 6.25% positivity rate for aSyn in the colon of patients diagnosed with MSA [173], a 40% positivity rate for the stomach, and an 8% rate for the colon, which was similar to the rates found in their PD patients and HCs [179]. One study involving patients with AD showed positive immunostaining for aSyn in the colon from 52% of patients, which was similar to the frequency found in HCs [174].

### 6.3. Isolated REM Sleep Behaviour Disorder

Sprenger et al. [191] reported positive immunostaining for phosphorylated aSyn in the colon in 23.5% of patients with iRBD, 5.3% of PD patients, and none of their controls, and Leclair-Visonneau et al. [192] found positive results in 64.4% of patients with PD and RBD compared with 13.3% in patients with PD and without RBD.

A recent study involving 94 PD patients, 72 iRBD patients, and 51 HCs showed significantly higher levels of aSyn aggregates in the stool of iRBD patients compared with PD patients and HCs, while PD patients showed non-significant differences when compared with HCs [193].

### 6.4. Other Diseases

Finally, colonic aSyn has been proposed as a marker for colorectal cancer, since hypermethylation of the SNCA gene was found to be significantly higher in patients with this disease than in healthy controls [194].

## 7. Studies Addressing aSyn Levels in the Skin

### 7.1. Parkinson’s Disease

The first description of the presence of aSyn in the skin from 3 of 16 PD patients and in 1 of 5 HCs was carried out by Michel et al. in 2005 [123]. Since then, there have been many reports regarding the detection of aSyn and phosphorylated aSyn in skin biopsies as a possible marker for PD and other synucleinopathies, which are summarized in Table 3 [123,168,195,196,197,198,199,200,201,202,203,204,205,206,207,208,209,210,211,212,213,214,215,216,217,218,219,220,221,222,223,224,225,226,227,228,229,230,231,232,233,234,235,236,237]. Depending on the method used and the site on which the skin biopsy was performed (with higher rates of positivity for cervical skin), the frequency of detection of aSyn in PD patients has shown great variability, ranging from positive percentages of lower than 10% [196,200] to 80–100% [201,203,208,214,216,217,219,220,221,223,224,225,226,227,228,229,231,232,234,235,236,237]. The frequency of immunopositivity for aSyn and/or aSyn deposition was evenly and significantly higher for PD patients than for HCs [195,197,198,199,201,202,203,204,206,207,209,210,212,223,226,227,228,229,230,231,232,234,235,236,237], and for patients with tauopathies (PSP, CBD, FTD) [168,198,202,204,212,218,230], non-specified type of atypical parkinsonism [199,223], and vascular parkinsonism [198]. Moreover, the positivity rate for aSyn of phosphorylated aSyn for HCs in many studies was 0% [199,202,203,207,210,212,213,214,215,216,219,220,222,223,229,231,232,234,236,237].

PD patients with RBD and patients with iRBD have shown a significantly higher rate of positivity for aSyn than those with PD without RBD [233]. PD patients with orthostatic hypotension have shown significantly higher immunostaining for phosphorylated aSyn and more widespread involvement of cholinergic and adrenergic skin nerve fibers than those without orthostatic hypotension [211].

Studies involving patients with genetic mutations associated with familial PD have shown the presence of aSyn deposition in 100% of patients carrying *SNCA* mutations [215,227], 66.7–100% of PD carrying *LRRK2* mutations [222,227], 60–83% of patients with *GBA1* gene mutations [213,227], and 0% of patients with *PRKN* mutations [198,227].

### 7.2. Other Synucleinopathies

Although initial studies failed to find a deposition of phosphorylated aSyn in the skin from patients with MSA [200,201], other studies found similar or slightly lower rates in MSA-P patients [202,210,214,217,218,221,230,232,236,237], or higher rates of positivity for aSyn or phosphorylated aSyn in MSA-P patients [168,225] than those found in patients with PD. MSA-P patients showed a higher affectation of somatic skin fibers while PD patients showed a predominant affection for autonomic skin fibers [217,225,232,236].

Similarly, patients diagnosed with LBD showed similar or slightly higher rates of positivity for aSyn or phosphorylated aSyn in the skin than those of PD patients [218,221,224,226,238], with some exceptions [212].

Finally, after the first description of the presence of phosphorylated aSyn in the skin nerve fibers from a patient diagnosed with pure autonomic failure (PAF) [239], several studies involving a short series of patients diagnosed with this disease showed positive immunostaining for phosphorylated aSyn in the skin in 100% [203,221,240].

### 7.3. Isolated REM Sleep Behaviour Disorder

The positivity rates of aSyn or phosphorylated aSyn in the skin for patients with iRBD ranged from 64 to 100% [168,223,229,233,235,241,242,243,244]. This is not surprising since many patients initially diagnosed with iRBD undergo phenoconversion to synucleinopathies, mainly PD and DLB, several years after the initial diagnosis [245,246,247].

### 7.4. Other Neurological Diseases

Donadio et al. [238] did not find phosphorylated aSyn deposits in 13 patients diagnosed with young-onset AD. Mejía et al. [119] reported lower aSyn levels in the skin, assessed by immunohistochemistry and flow cytometry, from 8 patients with relapsing–remitting multiple sclerosis during the relapse phase, compared with another 15 patients during remission and with 34 HCs. Levine et al. [248] described the positivity of phosphorylated aSyn in the skin from 7 of 22 (31.8%) patients with neuropathic postural tachycardia.

### 7.5. Other Medical Conditions

It has been described that human melanoma cell lines have a high expression of aSyn, with this protein being undetectable in non-melanoma skin cancer cell lines. In this regard, Matsuo et al. [249] reported the detection of aSyn in 86% of patients with primary and 85% of patients with metastatic melanoma, in 89% of nevus patients, and 0% in non-melanocytic cutaneous carcinoma and normal skin. Moreover, another group reported a significantly higher percentage of cells staining for aSyn in the skin from patients with melanoma (13.6%), than in those with nevi (7.7%), PD (3.3%), and HCs (1%) [209]. Studies on human melanoma cell lines have shown the presence of Ser19-phosphorylated aSyn in dot-like structures at the cell surfaces and the extracellular space by immunofluocesecence microscopy, and the release of microvesicles with Ser19-phosphorylated aSyn located in the vesicle membranes by immune-electron microscopy [250].

## 8. Studies Addressing Urinary aSyn Levels

Giri et al. [251] suggested the possible usefulness of determinations of aSyn in urine as a potential early biomarker in PD, with the collection of urine being non-invasive. Nam et al. [252] developed and validated an enzyme-linked immuno-absorbent assay (ELISA) to detect oligomeric aSyn in urine and showed a lack of differences in total aSyn between PD patients and controls, but they found a higher level of oligomeric aSyn recognized by MJFR-14-6-5-2 and lower levels of distinct oligomeric aSyn detected by ASyO5 in PD patients.

## 9. Discussion, Conclusions and Future Directions

Many clinical and experimental data suggest that the pathogenesis processes leading to PD begin several years before the onset of motor symptoms and clinical diagnosis. For this reason, the development of reliable biomarkers (included those related to aSyn) for the early detection of this disease and its differentiation from other synucleinopathies or other neurological diseases, represents an important opportunity to try treatments that could modify the course of the disease [253].

Data from studies addressing the serum/plasma levels of aSyn in PD patients compared with healthy controls have shown great variability, although the results of a meta-analysis showed a significant increase in the total, but not in oligomeric aSyn [68], and several studies showed increased levels of phosphorylated aSyn [22,25,66]. The results of studies comparing the serum/plasma aSyn levels of patients diagnosed with MSA, DLB, and PDD have been inconclusive, and studies comparing this value between patients with PSP, CBD, or frontotemporal dementia with that of controls showed non-significant differences. Studies on the serum/plasma levels of aSyn in patients with AD, autism spectrum disorders, and depression were controversial, while patients with Huntington’s disease and refractory epilepsy showed increased, and patients with multiple sclerosis and schizophrenia showed decreased, values of serum/plasma aSyn levels. Patients with attention deficit hyperactivity disorders have shown normal serum/plasma aSyn values. Finally, serum/plasma aSyn levels seem to be related to insulin resistance, and to be increased by exposure to alcohol, cocaine, and pesticides. The possible relevance of the determination of serum/plasma aSyn antibodies as reliable biomarkers for PD has been questioned because the great intra- and inter-cohort variability in the levels of this variable (up to 100-fold variation within groups) and the lack of a clear ability to discriminate PD patients from patients with other neurodegenerative diseases [64].

Studies of aSyn levels in erythrocytes have shown evenly increased values of total, oligomeric, and phosphorylated aSyn in PD and MSA patients compared with controls, decreased values in patients with AD, and variable results in patients with DLB.

Determinations of total, oligomeric, phosphorylated, and nitrated aSyn in the submandibular glands and minor salivary glands (using sufficient biopsy material) have shown significantly higher positivity rates in patients with PD ranging from 56% to 100%, than controls, although one study showed 39.9% positivity in controls as well. The possible value of salivary aSyn levels as a marker for PD has been analyzed in a meta-analysis that showed a significant decrease in the total salivary aSyn and a significant increase in the oligomeric aSyn and oligomeric/total aSyn ratio in patients with PD compared with controls [151], although the rate of detection of aSyn in saliva using RTQuIC was significantly higher in patients with PD. The positivity rate of the detection of aSyn in saliva by RTQuIC was found to increase in patients with MSA. Salivary aSyn levels were found to decrease in patients with AD and autism spectrum disorder. Patients with iRBD have shown significantly higher rates of positivity for aSyn in submandibular glands and minor salivary glands and higher rates of detection of aSyn in saliva using RTQuIC compared to controls.

Although the results of studies of nasal mucosa in biopsies and in autopsied patients were not useful for the detection of aSyn, further studies using RTQuIC have shown higher rates of positivity for aSyn in the nasal mucosa from patients with MSA, DLB and, to a lesser extent, in patients with PD compared with controls, although the rate of positivity could be increased with a more adequate selection of the area to obtain the biopsy sample.

The determinations of aSyn or phosphorylated aSyn in the colon have shown great variability among different studies comparing PD patients with controls (in many of them, the rate of positive was higher in PD patients, while in others there were no significant differences) with moderate rates of sensitivity and relatively high rates of specificity according to the results of a meta-analysis [189]. Similarly, the results of determinations of aSyn or phosphorylated aSyn in the stomach, duodenum, and small intestine have shown higher rates of positivity in PD patients than in controls, although these are restricted to a small number of studies. Studies on determinations of aSyn in MSA and in iRBD are scarce and insufficient to draw solid conclusions. The detection of aSyn aggregates in the stool could be a promising marker for iRBD according to a recent study [193].

In general, determinations of aSyn and phosphorylated aSyn in skin biopsies have shown a high rate of positivity in patients with PD and with other synucleinopathies such as MSA-P, DLB, and PAF, which permits the differentiation of these diseases from tauopathies, other atypical parkinsonisms, vascular parkinsonism, and controls. The determination of aSyn in skin biopsies seems to be useful as a marker of melanoma.

In summary, the results of determinations of aSyn, oligomeric aSyn, and phosphorylated aSyn in peripheral tissues could be potentially useful (combined with the studies of other potential biomarkers) and contribute to the discrimination of PD from controls and other causes of parkinsonism. However, the great variability between studies, specially in those performed in serum/plasma, should be considered, as this could be related to the method of study or to the kit used to measure aSyn levels. In addition, as described in previous sections, aSyn levels could be altered in several medical conditions that should be considered when selecting patients and controls. The most useful tissues for this purpose, according to the results of current studies, may be platelets, skin, and digestive tract.

We suggest that future studies trying to establish the possible role of the determination of aSyn in peripheral tissues as a marker for PD and other parkinsonisms should fulfil the following conditions:(1)Design including prospective and multicentre studies with a long-term follow-up period.(2)Recruitment of a large number of patients diagnosed with PD, other synucleinopathies, tauopathies, iRBD, and healthy controls. Subjects with other diseases that could influence the results regarding the concentration of aSyn in peripheral tissues (see previous sections) should not be chosen as healthy controls.(3)Collection of plasma/serum and erythrocytes, and biopsy specimens from salivary glands, nasal mucosa (preferably obtained from agger nasi), gut, and skin.(4)Analysis of samples with the best methodology and highest available uniformity.

## Figures and Tables

**Table 1 biomolecules-13-01263-t001:** Studies addressing serum and or plasma alpha-synuclein (aSyn) levels in patients with Parkinson’s disease (PD) and healthy controls (HCs). bSyn beta-synuclein; IFN interferon; IL interleukin; IPD idiopathic Parkinson’s disease; LRRK2 Leucine-rich repeat kinase 2; MSA multiple system atrophy; NLRP3 nod-like receptor protein 3; PGID: postural instability and gait difficulty; SIRT-2 sirtuin-2; *SNCA alpha-synuclein* gene; TD tremor dominant; TNF tumour necrosis factor.

Author, Year [Ref]	Study Subjects	Method	Main Findings
Woulfe et al., 2002 [9]	28 PD patients and 19 HCs	Two-step sandwich ELISA (Zymed, San Francisco, CA, USA)	Non-significant differences in serum aSyn antibody levels between PD patients and HCs.
Lee et al., 2002 [10]	105 PD patients and 51 HCs	ELISA kit (Amershan Biosciences, Slough, UK)	Increased plasma aSyn levels in PD patients.
Shi et al., 2010 [11]	126 PD patients and 122 HCs	Luminex Assays (Qiagen, Venlo, The Netherlands)	Non-significant trend towards low plasma aSyn levels in patients with PD (especially in those with stage IV).
Mata et al., 2010 [12]	86 PD patients and 78 HCs	Luminex Assays (Qiagen)	Non-significant differences in plasma aSyn levels between PD patients and HC.
Yanamandra et al. [13]	39 PD patients (27 early-onset) and 23 HCs	ELISA (Costar, Washington, D.C., USA) and immunoblot detection methods	Increase in serum autoantibody titers to aSyn oligomers, which decreased with PD progression.
Papachroni et al., 2011 [14]	31 PD patients and 26 HCs	Immunoblot analysis (Santa Cruz Biotechnology, Santa Cruz, CA, USA)	Similar frequency of the presence of serum aSyn antibodies in the patients with sporadic forms of PD and in controls. A total of 90% of patients with familial PD showed aSyn antibodies in serum.
Gruden et al., 2012 [15]	32 PD patients and 26 HCs	ELISA (Costar)	Increase in serum autoantibody titers to aSyn monomers, toxic oligomers, or fibrils in PD patients, which was associated with boosted levels of the pro-inflammatory cytokine IL-6 and TNF-α, but a decrease in IFN-γ concentration.
Bryan et al., 2012 [16]	30 PD patients and 14 HCs	Electroanalytical assays	Significant increase in serum aSyn levels in PD patients.
Smith et al. 2012 [17]	14 PD patients and 9 HCs	ELISA (AnaSpec, Inc., Fremont, CA, USA)	Non-significant differences in serum aSyn and aSyn antibodies levels between PD patients and HCs.
Hu et al., 2012 [18]	110 PD patients and 136 HCs. Two polymorphic variants of *SNCA* (Rep1 and rs11931074) were also studied	ELISA (Invitrogen, Waltham, MA, USA)	Non-significant differences in serum aSyn levels between PD patients and HCs. Increased frequency of rs11931074T allele in PD patients correlated with decreased serum aSyn.
Gorostidi et al., 2012 [19]	124 IPD patients, 32 PD patients carrying *LRRK2* mutations, and 109 HCs	ELISA (Pierce Biotechnology, Santa Cruz, CA, USA)	Significant decrease in plasma aSyn levels in patients with iPD, but not in *LRRK2* carriers, when compared with HCs.
Besong-Agbo et al., 2013 [20]	62 PD patients and 46 HCs	ELISA (Santa Cruz Biotechnology, Santa Cruz, CA, USA)	Serum antibodies against aSyn are significantly lower in PD patients than in HCs.
Caranci et al., 2013 [21]	69 PD patients and 110 HCs	ELISA (Invitrogen, Waltham, MA, USA)	Non-significant differences in plasma aSyn levels between PD patients and HCs. Plasma aSyn levels were associated with cognitive impairment, hallucinations, and sleep disorders in men.
Foulds et al., 2013 [22]	189 PD patients and 91 HCs	Immunoassay methods (Santa Cruz Biotechnology, Santa Cruz, CA, USA)	Non-significant differences in plasma total aSyn levels, but significantly higher plasma-phosphorylated aSyn levels in PD patients compared with HCs. Plasma total, but not plasma-phosphorylated aSyn, increased with time for up to 20 years after the initial symptoms.
Fernández et al., 2013 [23]	54 PD patients and 40 HCs	ELISA and immunoblot detection (BlueGene Biotech, Shangai, China)	Non-significant differences in serum aSyn levels between PD patients and HCs. Enhanced ratios between aSyn and nitrosylated aSyn in PD patients.
Maetzler et al., 2014 [24]	62 PD patients and 194 HCs	ELISA (Mediagnost, Reutlingen, Germany)	Non-significant differences in serum aSyn autoantibodies levels between PD patients and HCs.
Alvarez-Castelao et al., 2014 [25]	59 IPD patients, 104 carriers of *LRRK2* mutations (53 symptomatic), and 83 HCs	ELISA (Maxisorp; Nunc, Roskilde, Denmark)	Non-significant differences in serum aSyn antibody levels in patients with PD, *LRRK2* carriers, and HCs.
Xu et al., 2014 [26]	60 PD patients and 29 HCs	Impedimetric assay	Significant increase in serum levels of antibodies against aSyn in PD patients.
Shi et al., 2014 [27]	267 PD patients and 215 HCs	Luminex assay on exosome from plasma	Significant increase in plasma exosome aSyn levels in PD patients.
Heinzel et al., 2014 [28]	66 PD patients and 69 HCs	ELISA (Octapharma, Lachen, Switzerland)	Non-significant differences in serum aSyn autoantibodies levels between PD patients and HCs.
Gupta et al., 2015 [29]	97 PD patients and 97 HCs	ELISA (Invitrogen, Waltham, MA, USA)	Non-significant differences in serum aSyn levels between PD patients and HCs.
Bu et al., 2015 [30]	131 PD patients and 141 HCs	ELISA (Invitrogen, Waltham, MA, USA)	Significant increase in serum aSyn levels in PD patients, especially in those with an infectious bacterial and viral burden.
Caggiu et al., 2016 [31]	40 PD patients and 40 HCs	ELISA (LifeTein, South Plainfield, NJ, USA)	Significant increase in serum aSyn levels in PD patients, specially in those with an infectious viral burden.
Emelyanov et al., 2016 [32]	18 PD patients and 17 HCs	ELISA (Invitrogen, Waltham, MA, USA)	Non-significant differences in plasma aSyn levels between PD patients and HCs.
Ding et al., 2017 [33]	73 PD patients and 26 HCs	ELISA (Senbeijia, Nanjing, China)	Significantly higher plasma total aSyn in PD patients, which was higher in PGID than in TD patients.
Brudeck et al., 2017 [34]	46 PD patients and 41 HCs	ELISA and Meso-Scale Discovery (MSD) electro-chemiluminescence assays (MULTI-ARRAY Assay Systems, Rockville, MD, USA)	Significant decrease in plasma levels of antibodies against aSyn and phosphorylated aSyn in PD patients.
Horvath et al., 2017 [35]	46 PD patients and 30 HCs	ELISA (not specified)	Significant increase in plasma levels of antibodies against aSyn in PD patients.
Shalash et al., 2017 [36]	46 PD patients and 20 HCs	ELISA (MyBioSource Inc., San Diego, CA, USA)	Significant increase in serum levels of autoantibodies against aSyn in PD patients.
Goldman et al., 2018 [37]	115 PD patients and 88 HCs	ELISA (BioLegend, San Diego, CA, USA)	Non-significant differences in plasma aSyn levels between PD patients and HCs.
Singh et al., 2018 [38]	38 PD patients and 33 HCs	Real-time label-free surface plasmon resonance (SPR) with BIAcore-3000 (Wipro GE Healthcare, Upsala, Sweden)	Significant increase in serum aSyn levels in PD patients, which was negatively correlated with decreased serum mortalin levels.
Malec-Litwinowicz et al., 2018 [39]	58 PD patients and 38 HCs	ELISA (Invitrogen, Waltham, MA, USA)	Non-significant differences in plasma aSyn levels between PD patients and HCs.
Akhtar et al., 2018 [40]	53 PD patients and 16 HCs	ELISA (Thermo Scientific, Waltham, MA, USA)	Non-significant differences in serum aSyn levels between PD patients and HCs.
Ng et al., 2019 [41]	170 PD patients and 51 HCs	Ultrasensitive single-molecular array (Simoa, Quanterix, MA, USA),	Significantly higher plasma aSyn levels in PD patients.
Singh et al., 2019 [42]	68 PD patients and 68 HCs	Real-time label-free surface plasmon-resonance (SPR) with BIAcore-3000 (Wipro GE Healthcare, Upsala, Sweden)	Significant increase in serum aSyn levels in PD patients, which was correlated with increased serum sirtuin 2 (SIRT2) levels.
Folke et al., 2019 [43]	43 PD patients and 59 HCs	ELISA setups developed in-house	Non-significant differences in plasma naturally occurring antibody levels against aSyn between PD patients and HCs.
Lin et al., 2019 [44]	122 PD patients and 68 HCs	Immunomagnetic reduction (IMR)-based immunoassay (MagQu, Taipei, Taiwan)	Significantly higher total plasma and Ser129-phosphorylated aSyn levels in PD patients.
Chen et al., 2019 [45]	30 PD patients and 30 HCs	ELISA, double-antibody sandwich method (American Type Culture Collection, ATCC, Manassas, VA, USA).	Significant increase in plasma aSyn oligomer levels in PD patients.
Wang et al., 2019 [46]	45 PD patients and 45 HCs	ELISA (Santa Cruz Biotechnology, Santa Cruz, CA, USA)	Significant increase in plasma aSyn levels in PD patients.
Wang et al., 2019 [47]	59 PD patients and 60 HCs	Immunomagnetic reduction (IMR)-based immunoassay (MagQu, Taipei, Taiwan)	Significant increase in serum aSyn levels in PD patients, which was correlated with increased serum Rab35 levels.
Chatterjee et al., 2020 [48]	27 PD patients and 15 HCs	ELISA (Thermofisher Scientific, Waltham, MA, USA, and My Biosource, San Diego, CA, USA)	Significant increase in serum aSyn levels in late-onset PD patients, which was correlated with increased serum cytosolic nod-like receptor protein 3 (NLRP3) levels.
Chen et al., 2020 [49]	42 PD patients with normal cognition and 12 HCs	Immunomagnetic reduction (IMR)-based immunoassay (MagQu, Taipei, Taiwan)	Significant increase in plasma aSyn levels in PD patients.
Fan et al., 2020 [50]	43 early PD and 24 HCs	Meso-Scale Discovery (MSD) electro-chemiluminescence assay (Rockville, MD, USA)	Significant increase in plasma aSyn levels in PD patients. Positive correlation between aSyn levels and severity of PD.
Chang et al., 2020 [51]	48 PD patients and 40 HCs	Immunomagnetic reduction (IMR)-based immunoassay (MagQu, Taipei, Taiwan)	Significant increase in serum and plasma aSyn levels in PD patients.
Wang et al., 2020 [52]	40 PD patients and 40 HCs	ELISA (Santa Cruz Biotechnology, Santa Cruz, CA, USA)	Significantly higher plasma oligomeric and phosphorylated aSyn levels in PD patients.
Bougea et al., 2020 [53]	30 PD patients and 30 HCs	ELISA (Fujirebio, Gent, Belgium)	Significant increase in serum and similar plasma aSyn levels in PD patients compared with HCs.
Emmanouili-dou et al., 2020 [54]	153 PD patients (124 genetically undetermined -GUPD—and 29 A53T mutation carriers—A53T-PD), and 97 HCs	ELISA (in-house)	Significant increase in serum (but not plasma) aSyn levels in GUPD patients, and significant decrease in serum and plasma aSyn levels in A53TPD patients. Modest negative correlation between aSyn levels and UPDRS motor subscale score.
Wijeyekoon et al., 2020 [55]	41 PD patients and 41 HCs	Meso-Scale Discovery (MSD) electro-chemiluminescence assay (Rockville, MD, USA)	Significant decrease in serum and plasma aSyn levels in PD patients.
Chahine et al., 2020 [56]	59 PD patients and 21 HCs	ELISA (BioLegend, San Diego, CA, USA)	Non-significant differences in plasma total aSyn levels between PD patients and controls.
Lin et al., 2020 [57]	57 PD patients with normal cognition and 97 HCs	Immunomagnetic reduction (IMR)-based immunoassay (MagQu, Taipei, Taiwan)	Significantly higher plasma total aSyn levels in PD patients.
Shim et al., 2020 [58]	20 PD patients and 20 HCs	ELISA (Abcam, in-house)	Significantly lower plasma total aSyn levels in PD patients.
Nasirzadeh et al., 2021 [59]	20 PD patients and 20 HCs	ELISA (Shanghai Crystal Day Biotech Co., Ltd.).	Significantly higher serum oligomeric aSyn levels in PD patients.
Stuendl et al., 2021 [60]	96 PD patients and 41 HCs	Meso-Scale Discovery (MSD) electro-chemiluminescence assay (Gaithersburg, MD, USA)	Non-significant differences in plasma total aSyn levels between PD patients and controls. Significantly higher extravesicular aSyn levels in PD patients than in controls.
Jiang et al., 2021 [61]	290 PD patients and 191 HCs	Meso-Scale Discovery (MSD) electro-chemiluminescence assay (Gaithersburg, MD, USA)	Significant increase in aSyn levels in L1CAM-immunocaptured serum exosomes compared with HCs.
Schulz et al., 2021 [62]	151 PD patients and 20 HCs	ELISA (BioLegend, San Diego, CA, USA)	Non-significant differences in serum aSyn levels between PD patients and HCs.
Ghit and El Deeb, 2022 [63]	20 PD patients and 15 HCs	ELISA (Biomatik, Kitchener, ON, Canada)	Significant increase in serum aSyn levels in PD patients compared with controls.
Garg et al., 2022 [64]	157 PD patients 46 HCs, and 92 patients with other neuro-degenerative disorders	ELISA (in-house)	Significant decrease in serum aSyn and bSyn autoantibodies levels in patients with PD and other neurodegenerative diseases (a mix of AD, AD-related dementia, tauopathies, and “other movement disorders”) compared with HCs.
Chatterjee et al., 2022 [65]	60 PD patients (24 tremor-dominant and 36 with gait difficulties and postural instability)	ELISA (Thermofisher Scientific, Waltham, MA, USA, and My Biosource, San Diego, CA, USA)	Similar total serum aSyn levels and significantly higher serum 129Ser-phosphorylated aSyn levels in patients with GPID compared with TD PD patients.
Chen et al., 2022 [66]	10 PD patients and 11 HCs	Real-time label-free surface plasmon resonance (SPR)	Significantly higher serum 129Ser-phosphorylated aSyn levels in PD patients.
Youssef et al., 2023 [67]	29 PD patients and 30 HCs	Meso-Scale Discovery (MSD) electro-chemiluminescence assay (Gaithersburg, MD, USA)	Non-significant differences in serum aSyn levels between PD patients and HCs.

**Table 2 biomolecules-13-01263-t002:** Studies addressing serum and or plasma alpha-synuclein (aSyn) levels in patients with other parkinsonian syndromes and healthy controls (HCs). CBD corticobasal degeneration; DLB dementia with Lewy bodies; MSA multiple system atrophy; PDD Parkinson’s disease dementia, PD-MCI Parkinson’s disease with mild cognitive impairment; PSP progressive supranuclear palsy; RBD REM sleep behaviour disorder; SIRT-2 sirtuin-2.

Author, Year [Ref]	Study Subjects	Method	Main Findings
Lee et al., 2002 [10]	31 MSA patients and 51 HCs	ELISA kit (Amershan Biosciences, Slough, UK)	Increased plasma aSyn levels in MSA patients (but less than PD patients of the same series).
Brudeck et al., 2017 [34]	18 MSA patients and 41 HCs	ELISA and Meso-Scale Discovery (MSD) electro-chemiluminescence assays (MULTI-ARRAY Assay Systems, Rockville, MD, USA)	Significant decrease in plasma levels of antibodies against aSyn and phosphorylated aSyn in MSA patients.
Singh et al., 2019 [42]	34 MSA + PSP patients and 68 HCs	Real-time label-free surface plasmon resonance (SPR) with BIAcore-3000 (Wipro GE Healthcare, Upsala, Sweden)	Significant increase in serum aSyn levels in AMS + PSP patients (although less than that of 68 PD patients in the same study), which was correlated with increased serum sirtuin 2 (SIRT2) levels.
Folke et al., 2019 [43]	34 MSA patients and 59 HCs	ELISA setups developed in-house	Non-significant differences in plasma naturally occurring antibody levels against aSyn between MSA patients and HCs.
Wang et al., 2019 [47]	20 MSA patients and 60 HCs	Immunomagnetic reduction (IMR)-based immunoassay (MagQu, Taipei, Taiwan)	Non-significant differences in serum aSyn levels between MSA patients and HCs.
Jiang et al., 2021 [61]	50 MSA patients and 191 HCs	Meso-Scale Discovery (MSD) electro-chemiluminescence assay (Gaithersburg, MD, USA)	Non-significant differences in aSyn levels in L1CAM-immunocaptured serum exosomes between MSA patients and HCs.
Schulz et al., 2021 [62]	17 MSA patients and 20 HCs	ELISA (BioLegend, San Diego, CA, USA)	Non-significant differences in serum aSyn levels between MSA patients and HCs.
Wang et al., 2019 [47]	19 PSP patients and 60 HCs	Immunomagnetic reduction (IMR)-based immunoassay (MagQu, Taipei, Taiwan)	Non-significant differences in serum aSyn levels between PSP patients and HCs.
Stuendl et al., 2021 [60]	50 PSP patients and 41 HCs	Meso-Scale Discovery (MSD) electro-chemiluminescence assay (Gaithersburg, MD, USA)	Non-significant differences in plasma total aSyn levels and extravesicular aSyn levels in PSP patients and HCs.
Jiang et al., 2021 [61]	116 PSP patients and 191 HCs	Meso-Scale Discovery (MSD) electro-chemiluminescence assay (Gaithersburg, MD, USA)	Non-significant differences in aSyn levels in L1CAM-immunocaptured serum exosomes between MSA patients and HCs.
Schulz et al., 2021 [62]	38 PSP patients and 20 HCs	ELISA (BioLegend, San Diego, CA, USA)	Non-significant differences in serum aSyn levels between PSP patients and HCs.
Maetzler et al., 2011 [69]	14 DLB patients and 31 HCs		Significantly higher serum autoantibody levels against aSyn in patients with DLB.
Laske et al., 2011 [70]	40 DLB and 40 HCs	ELISA (Invitrogen, Waltham, MA, USA)	Significantly lower aSyn levels in patients with DLB than in HCs.
Koehler et al., 2013 [71]	19 DLB and 16 HCs	ELISA (in-house)	Significantly higher serum autoantibody levels against aSyn in patients with DLB than in HCs.
Bougea et al., 2020 [53]	29 DLB patients and 30 HCs	ELISA (Fujirebio, Gent, Belgium)	Significant increase in serum and plasma aSyn levels in DLB patients compared with HCs.
Stuendl et al., 2021 [60]	50 DLB patients and 41 HCs	Meso-Scale Discovery (MSD) electro-chemiluminescence assay (Gaithersburg, MD, USA)	Non-significant differences in plasma total aSyn levels and extravesicular aSyn levels in DLB patients and HCs.
Schulz et al., 2021 [62]	45 DLB patients and 20 HCs	ELISA (BioLegend, San Diego, CA, USA)	Non-significant differences in serum aSyn levels between DLB patients and HCs (although DLB patients had lower serum aSyn levels than PD patients from the same cohort).
Maetzler et al., 2011 [69]	13 PDD patients and 31 HCs	ELISA (Corning, Lowell, MA, USA)	Significantly higher serum autoantibody levels against aSyn in patients with PDD.
Maetzler et al., 2014 [24]	31 PDD patients and 194 HCs	ELISA (Mediagnost, Reutlingen, Germany)	Non-significant differences in serum aSyn antibody levels between PDD patients and HCs.
Chen et al., 2020 [49]	50 PDD patients, 66 PD-MCI, and 12 HCs	Immunomagnetic reduction (IMR)-based immunoassay (MagQu, Taipei, Taiwan)	Significant increase in plasma aSyn levels in PDD and PD-MCI patients.
Bougea et al., 2020 [53]	18 PDD patients and 30 HCs	ELISA (Fujirebio, Gent, Belgium)	Significant increase in serum and plasma aSyn levels in PDD patients compared with HCs.
Wang et al., 2020 [72]	61 PDD patients and 50 HCs	ELISA (in-house)	Significant decrease in serum levels of antibodies against aSyn in PDD patients compared with HCs.
Lin et al., 2020 [57]	87 PDD patients, 66 PD-MCI, and 97 HCs	Immunomagnetic reduction (IMR)-based immunoassay (MagQu, Taipei, Taiwan)	Significantly higher plasma total aSyn in PDD and PD-MCI patients than in controls and PD patients with normal cognition from the same group.
Kronimus et al., 2016 [73]	18 PDD patients and 18 PDND patients	ELISA (Dianova GmbH, Hamburg, Germany)	Significantly lower aSyn levels in patients with PDD.
Jiang et al., 2021 [61]	88 CBD patients and 191 HCs	Meso-Scale Discovery (MSD) electro-chemiluminescence assay (Gaithersburg, MD, USA)	Non-significant differences in aSyn levels in L1CAM-immunocaptured serum exosomes between CBD patients and HCs.
Schulz et al., 2021 [62]	16 CBD patients and 20 HCs	ELISA (BioLegend, San Diego, CA, USA)	Non-significant differences in serum aSyn levels between CBD patients and HCs.
Hu et al. 2015 [74]	69 PD patients with RBD, 156 PD patients without RBD, and 21 HCs	ELISA (CUSABIO, Wuhan, China)	Increased serum aSyn levels in patients with RBD compared with those without RBD and with HCs.

**Table 3 biomolecules-13-01263-t003:** Studies addressing skin alpha-synuclein (aSyn) detection in patients with Parkinson’s disease and/or other parkinsonian syndromes, and healthy controls (HCs). AD Alzheimer’s disease; ALS amyotrophic lateral sclerosis; CBD corticobasal degeneration; DLB dementia with Lewy bodies; ET essential tremor; the glucocerebrosidase (GBA1); IPD idiopathic Parkinson’s disease; iRBD idiopathic or isolated REM sleep behaviour disorder; LRRK2 Leucine-rich repeat kinase 2; MSA multiple system atrophy; MSA-P multiple system atrophy-parkinsonism; MSA-C multiple system atrophy cerebellar; PAF pure autonomic failure; PARK2 or PRK parkin; PSP progressive supranuclear palsy; RBD REM sleep behaviour disorder; RT-QuIC real-time quaking-induced conversion; SIRT-2 sirtuin-2; TDP transactive response DNA-binding protein.

Author, Year [Ref]	Study Subjects	Main Findings
Michell et al., 2005 [123]	16 PD patients and 5 HCs	Detection of aSyn in the skin from 3 patients and 1 HC.
Ikemura et al., 2008 [195]	Prospective study with 279 autopsied patients (85 with LB pathology) Retrospective study with subclinical of clinical LB disease	Positive immunoreactivity to 129Ser-phosphorylated aSyn in the unmyelinated fibers of the dermis in 23.5% of patients with LB pathology and in none of 194 patients without LB pathology. Positive immunoreactivity to 129Ser-phosphorylated aSyn in 70% of PD patients with and without dementia, in 40% of patients with LB disease, and 0% of cases with MSA, PSP, and CBD.
Miki et al., 2010 [196]	20 PD patients	Positive immunoreactivity to 129Ser phosphorylated aSyn in 2 patients (10%).
Wang et al., 2013 [197]	20 PD patients and 14 HCs	Higher deposition of aSyn alone or normalized to nerve density fiber within pilomotor and sudomotor, but not sensory nerves, in PD patients compared with controls.
Donadio et al., 2014 [198]	21 PD patients, 20 patients with other parkinsonisms (10 vascular, 6 tauopathies, 4 with parkin mutations), and 30 HCs	Positive immunoreactivity to 129Ser-phosphorylated aSyn in the small nerve fibers of all PD patients and none of the groups of other parkinsonisms and HCs.
Rodríguez-Leyva et al., 2014 [199]	34 PD patients, 33 patients with atypical parkinsonism, and 26 HCs	Positive immunohistochemistry and immunofluorescence for aSyn in 58% of the cells in the spinous layer, 62% in the pilosebaceous unit, and 58% of the eccrine glands in PD patients, 7%, 7%, and 0%, respectively, in patients with atypical parkinsonisms, and no expression in the control group.
Haga et al., 2015 [200]	38 PD patients and 13 MSA patients	Presence of 129Ser-phosphorylated aSyn aggregates in the skin of 5.3% PD patients but none in patients with MSA.
Zange et al. 2015 [201]	10 PD patients, 10 MSA patients, and 6 ET patients	Deposition of 129Ser-phosphorylated aSyn in skin sympathetic nerve fibers in 100% of PD patients and none of the patients with MSA or ET.
Doppler et al., 2015 [202]	30 PD patients, 12 MSA patients, 15 patients with tauopathies, and 39 HCs	Positive immunofluorescence for 129Ser-phosphorylated aSyn in dermal nerves (basically in unmyelinated somatosensory fibers) from 67% of patients with PD and MSA and in none of the patients with tauopathies or controls.
Donadio et al., 2016 [203]	16 PD patients, 14 PAF patients, and 15 HCs	Positive immunostaining for 129Ser-phosphorylated aSyn in small nerve fibers from all PD and PAF patients and in none of the HCs. Native aSyn was similarly expressed in the 3 groups. aSyn deposits were found in all analyzed skin samples from PAF but only in 49% of PD patients.
Rodríguez-Leyva et al., 2016 [204]	17 PD patients, 10 PSP patients, and 17 HCs	Significantly higher immunopositivity for aSyn in PD patients compared with PSP patients and HCs.
Gibbons et al., 2016 [205]	28 PD patients (15 with autonomic failure) and 23 HCs	Significantly higher immunopositivity for aSyn and aSyn normalized for sympathetic nerve fibers in PD patients (both with and without autonomic failure) compared with HCs.
Gibbons et al., 2017 [206]	11 PD patients and 5 non-synucleinopathy controls (all post-mortem)	Significantly higher aSyn deposition in pilomotor, sudomotor, and vasomotor nerves in PD patients than in controls. This was not correlated with age, duration of PD, or severity of PD.
Doppler et al., 2017 [207]	25 early PD patients, 18 RBD patients, and 20 HCs	Positive immunofluorescence staining for 129Ser-phosphorylated aSyn in 80% of early PD patients, 55.6% of RBD patients, and none of the HCs.
Donadio et al., 2017 [208]	28 IPD patients (15 with unilateral and 13 with bilateral symptoms)	Positive immunostaining for 129Ser-phosphorylated aSyn in the affected motor side in 20%, in both sides in 60%, and in the nonaffected motor side in 20% of patients with unilateral symptoms at the cervical paraspinal region. Positivity in 100% of PD patients at the cervical, and 62% in the thoracic paraspinal region.
Rodríguez-Leyva et al., 2017 [209]	8 PD patients and 9 HCs	Significantly higher percentage of immunopositivity for aSyn in the basal layer of the epidermis from PD patients compared with HCs (but lower than that of patients with nevi and melanoma included in the same study).
Donadio et al., 2018 [210]	15 iPD patients, 12 DLB patients, 5 PAF patients, 12 MSA patients (5 MSA-P and 7 MSA-C) and 10 HCs	All synucleinopathy patients (except 4 with MSA-C) and none of the HCs showed positive immunostaining for 129Ser-phosphorylated aSyn, although HCs showed a weak co-staining for native aSyn.
Donadio et al., 2018 [211]	28 PD patients (14 of them with orthostatic hypotension)	Significantly higher positive immunostaining for 129Ser-phosphorylated aSyn in PD patients with orthostatic hypotension, with widespread autonomic cholinergic and adrenergic skin nerve fibers’ involvement.
Melli et al., 2018 [212]	19 PD patients, 7 patients with possible alpha-synucleinopathy (5 MSA and 2 LBD), 6 with possible tauopathy (4 PSP and 2 CBD), and 17 HCs	Significantly higher positive immunostaining for 129Ser-phosphorylated aSyn in PD patients than for the groups of alpha-synucleinopathies, tauopathies, and controls.
Doppler et al., 2018 [213]	10 PD patients with different mutations in the *glucocerebrosidase* (*GBA1*) gene	Positive immunostaining for 129Ser-phosphorylated aSyn in 60% of patients, mainly detected in autonomic nerve fibers, but also in somatosensory fibers.
Kuzkina et al., 2019 [214]	27 early PD patients, 8 MSA-P patients, and 21 HCs	Positive immunostaining for 129Ser-phosphorylated aSyn (truncated and aggregated protein) in dermal nerve fibers from 85% of PD patients, 75% of MSA-P patients, and none of the controls.
Carmona-Abellán et al., 2019 [215]	7 E46K *SNCA* carriers (3 DLB, 2 PAF, 1 PD, 1 asymptomatic), 2 *PARK2* carriers and 2 HCs	All E46K *SNCA* carriers (especially those with PAF), and no *PARK2* carriers of HCs showed phosphorylated aSyn deposits in epidermal and dermal structures including nerve fascicles and glands.
Mazzeti et al., 2020 [216]	57 PD patients (19 with unaffected monozygotic twins) and 48 HCs (19 asymptomatic monozygotic twins of PD patients)	Significant increase in aSyn oligomers, determined by proximity ligation assay, in PD patients (82%) compared with controls (0% of HCs including twins).
Donadio et al., 2020 [217]	25 patients with PD and 25 with MSA-P, all of them with chronic orthostatic hypotension	Positive immunostaining for intraneural phosphorylated aSyn in 72% of MSA-P patients (most of them in somatic fibers or epidermic plexi, 12% in autonomic skin fibers) and in 100% of PD patients (most of them in autonomic skin fibers, 16% in somatic fibers).
Wang et al., 2020 [218]	57 patients with synucleinopathies (47 PD, 7 LBD, 3 MSA), 30 with tauopathies (17 AD, 8 PSP, 5 CBD), and 43 non-neurodegenerative controls (all samples from autopsies) 20 living patients with PD and 20 controls	RT-QuIC followed by protein misfolding cyclic amplification analysis showed higher rates of positivity for aSyn in patients with PD than in patients with LBD and MSA (non-significant differences) and patients with different tauopathies and controls (significant differences).
Liu et al., 2020 [219]	90 PD patients and 30 HCs	Positive immunofluorescence for phosphorylated aSyn in 83.3% of PD patients and 0% of HCs. The sensitivity was enhanced by using two biopsy sites (cervical/distal leg of two cervical sites).
Wang et al., 2020 [220]	29 PD patients and 21 HCs	Positive detection of phosphorylated aSyn of 100% for 50 µm sections, 90% for 20 µm sections, and 73% for 10 µm sections in PD patients, and lack of detection in HCs.
Giannoccaro et al., 2020 [221]	21 IPD, 7 DLB, 13 MSA, and 13 PAF patients	Positive detection of phosphorylated aSyn in 100% of DLB and PAF, 95.2% of IPD, and 69.2% of MSA (in MSA affectation of autonomic fibers is rare, with the affectation of somatic subepidermal plexus being more frequent).
Yang et al., 2021 [222]	59 PD patients (12 carrying the *LRRK2* G2385R variant) and 30 HCs	Positive immunofluorescence for phosphorylated aSyn in 70.2% of PD non-carriers, in 66.7% of PD carriers of the *LRRK2* G2385R variant, and in 0% of HCs.
Al-Qassabi et al., 2021 [223]	28 patients with Lewy body neuropathology and 23 controls (autopsied patients) 20 PD, 10 atypical parkinsonism, 28 iRBD, and 21 HCs (living patients)	Positive immunohistochemistry for phosphorylated aSyn in 92.9% of patients with Lewy body neuropathology and 0% of controls from autopsy samples. Positive immunohistochemistry for phosphorylated aSyn in 70% of PD patients, 20% of atypical Parkinsonism, 82% of iRBD patients, and 0% of controls.
Donadio et al., 2021 [224]	33 patients with synucleinopathies (17 PD, 5 DLB, 8 probable MSA, 3 PAF), and 38 patients with non-synucleino-pathies (15 AD, 6 vascular parkinsonism, 1 neuroleptic-induced parkinsonism. 2 vascular dementia, 7 tauopathies or TDP proteinopathy, 6 ALS), and 24 controls (mainly with peripheral neuropathies)	Mean skin aSyn thioflavin fluorescence reaction was significantly higher in patients with synucleinopathies than in patients with non-synucleinopathies and controls, with non-significant differences between patients with no-synucleinopathies and controls.
Brumberg et al., 2021 [225]	21 PD patients and 21 MSA patients	Positive immunofluorescence for phosphorylated aSyn in 47.6% of PD patients (mainly in autonomic structures) and 81% of MSA patients (mainly in somatosensory fibers).
Mammana et al., 2021 [226]	13 PD patients, 15 DLB patients, and 41 controls (living patients) 2 DLB/PD patients, 13 incidental Lewy body disease patients, and 40 non-Lewy body disease (autopsied patients)	Positive RT-QuIC for aSyn in 76.9%, 100%, and 4.9%, respectively. Positive RT-QuIC for aSyn in 100%, 85.7%, and 2.5%, respectively.
Isonaka et al., 2021 [227]	30 subjects with pathogenic mutations (3 *SNCA*, 10 *PRKN*, 7 *LRRK2*, 7 *GBA*, 3 *PARK7/DJ1*, 25 of them with PD), 19 patients iPF, and 16 HCs	Positive immunofluorescence for aSyn was above the control range in 100% of subjects with *SNCA* mutations, 100% with *LRRK2* mutations, 95% with idiopathic PD, 83% with *GBA* mutations, and 0% with biallelic *PRKN* mutations, although mean deposition of aSyn was significantly higher for the biallelic *PRNK* mutations than for HCs.
Kuzkina et al., 2021 [228]	34 PD patients and 30 HCs	Positive RT-QuIC for aSyn in 82.4% of PD patients (and 8.8% as intermediate) and 10% of HCs (and 3.3% as intermediate).
Miglis et al., 2021 [229]	28 PD patients, 25 iRBD patients, and 18 HCs	Positive immunofluorescence for phosphorylated aSyn in 96% of PD patients, 64% of iRBD patients, and 0% of controls.
Vacchi et al., 2021 [230]	30 PD patients, 13 MSA patients, 11 patients with tauopathies, and 22 HCs	Significantly higher percentage of positivity of aSyn determined by proximity ligation assay in PD (80%) and MSA (66.7%) patients (with greater area within nerves in MSA) than in patients with tauopathies (18.2%) and HCs (22.7%). Significantly higher percentage of positivity of immunofluorescence for aSyn and phosphorylated aSyn in PD and MSA than in tauopathies and HCs, but with less sensitivity.
Giannoccaro et al., 2022 [231]	26 PD patients, 18 PSP patients, 8 CBD patients, and 26 HCs	Positive immunofluorescence for phosphorylated aSyn in 100% of PD, 5.6% of PSP, 12.5% of CBD, and 0% of HCs.
Nolano et al., 2022 [232]	57 early PD, 43 MSA-P patients, and 100 skin biopsy controls	Positive immunofluorescence for phosphorylated aSyn in the skin nerves from 96% of PD patients and 91% of MSA-P patients and in 0% of skin biopsy controls. Phosphorylated aSyn deposits in autonomic nervous structures were significantly more frequent in PD than in MSA-P patients.
Doppler et al., 2022 [233]	43 PD patients (19 with RBD), and 43 iRBD patients	Positive immunohistochemistry for phosphorylated aSyn in 52.4% of PD patients without RBD, 81.8% of PD + RBD patients, and 79.1% of iRBD patients.
Oizumi et al., 2022 [234]	10 IPD living patients and 4 autopsied controls	Positive immunofluorescence for phosphorylated aSyn in dermal macrophages from 100% of PD patients and in 0% of controls.
Kuzkina et al., 2023 [235]	39 PD patients, 38 iRBD patients, and 23 HCs	Detection of aSyn aggregation and positive immunohistochemistry for phosphorylated aSyn, respectively, in 87.2% and 70% of PD patients, 97.4% and 78.4% of iRBD patients, and 13% and 7.9% of HCs.
Donadio et al., 2023 [236]	34 PD patients, 46 MSA patients (29 MSA-P and 17 MSA-C), 16 DLB patients, and 50 HCs	Positive immunofluorescence for phosphorylated aSyn in 100% of patients with PD and DLB, in 78% of MSA patients, and 0% of HCs. MSA was positive in somatic neurons and in Schwann cell cytoplasmic inclusions, and PD/DLB in autonomic neurons.
Gibbons et al., 2023 [237]	54 PD patients, 31 MSA patients, and 24 HCs	Positive immunofluorescence for phosphorylated aSyn in 94.4% of PD patients, 100% of MSA patients, and 0% of HCs. Patients with MSA showed greater aSyn deposition and more widespread peripheral distribution than patients with PD.
Kuzkina et al., 2023 [168]	27 PD patients, 18 iRBD patients, 3 MSA patients, and 3 PSP patients	Rates of positivity for seed amplification assay for aSyn of 78.9% for PD, 100% for iRBD and MSA, and 0% for PSP patients.

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
