# Peer review of "Alpha-Synuclein in Peripheral Tissues as a Possible Marker for Neurological Diseases and Other Medical Conditions"

_biomolecules, 2023, doi:10.3390/biom13081263_

Round 1

Reviewer 1 Report

The extensive literature review by Jimenez et al. focuses on reporting alpha-synuclein (aSyn) levels in different types of peripheral tissues, such as serum and/or plasma, which may be altered in various neurodegenerative disorders, particularly those characterized by neuronal or glial inclusions composed of aggregated aSyn. This review highlights that results are inconsistent across all serum/plasma studies and depend on the specific neurodegenerative condition being studied. On the other hand, the authors collected studies supporting that other neurological and psychiatric disorders that do not fall under the synucleinopathies umbrella, like Huntington's disease (HD) and certain types of epilepsy, demonstrate increased aSyn levels. The authors concluded that peripheral tissues such as platelets, skin, and the digestive tract may be potential sources for aSyn biomarkers in PD and other synucleinopathies. In addition, this review suggests that future studies need to fulfill specific conditions, including prospective and multicenter design, recruitment of many patients and controls, and use of standardized methodologies to further establish the usefulness of aSyn determinations in these tissues as biomarkers for neurodegenerative disorders.

Major Comments:

1) The authors need to address the fact that the kit or antibodies used to measure aSyn species add variability when comparing results from different studies. Therefore, the tables provided in the review will be more informative if the authors also include a column addressing the Technique or kit/antibody used to measure aSyn levels.

2) The authors limit themselves to reporting most of the time if there is an increase in aSyn levels but do not elaborate on whether this increase or decrease is associated with disease severity (e.g., motor symptoms severity or cognitive decline).

Minor Comments:

1) Although the manuscript reads well, some typos and sentences need clarification. I highlighted these in the uploaded pdf.

Author Response

REVIEWER # 1

Major Comments:

1) The authors need to address the fact that the kit or antibodies used to measure aSyn species add variability when comparing results from different studies. Therefore, the tables provided in the review will be more informative if the authors also include a column addressing the Technique or kit/antibody used to measure aSyn levels. 

  1. We have added a column in tables 1 and 2 specifying the methods used to measure aSyn levels.

2) The authors limit themselves to reporting most of the time if there is an increase in aSyn levels but do not elaborate on whether this increase or decrease is associated with disease severity (e.g., motor symptoms severity or cognitive decline).

Ok. Although association between aSyn levels and other clinical and biochemical data were commented in Table 1, now we have added the following paragraph in the text:  “Several studies described an association between serum/plasma total aSyn and cognitive impairment, hallucinations and sleep disorders [21], a negative correlation between aSyn levels and PD severity [54], lower serum aSyn levels in patients with advanced PD [11, 13], and higher serum plasma total aSyn in PD patients with predominant postural instability and gait difficulty compared with those with tremor-dominant PD [33, 65], and in those patients with an infectious bacterial [30] and viral [30, 31] burden. Serum oligomeric aSyn levels  Serum/plasma aSyn levels have been found correlated with increased serum sirtuin 2 [42], Rab35 [47], nod-like receptor protein 3 [48] levels, and with decreased serum mortalin levels [38]”

3) Although the manuscript reads well, some typos and sentences need clarification. I highlighted these in the uploaded pdf. 

Ok, revised and corrected.

Reviewer 2 Report

Riview of a manuscript “ALPHA-SYNUCLEIN IN PERIPHERAL TISSUES AS A POSSIBLE MARKER FOR NEUROLOGICAL DISEASES” by Jiménez-Jiménez and coauthors submitted to “Biomolecules”

Synucleinopathies are a group of severe neurodegenerative diseases for which there is no efficient treatment altering the course of these disorders. The use of biomarkers to identify the early stages of these diseases is also limited. Analysis of alpha-synuclein in peripheral tissues, including blood cells, olfactory mucosa, digestive tract, skin, etc. may bring important data about the first steps of alterations. This will allow early identification and diagnosis of the disease and the beginning of specific treatment. The authors of this review summarize data from studies on alpha-synuclein concentration in various peripheral tissues in synucleinopathies, including Parkinson’s. This is an important area of biomedical research, and the data presented in the manuscript will be interesting for the readers of “Biomolecules”.

The following corrections and additions should be made.

 Abstract:

Line 24:”…to discriminate PD from controls and form other causes of parkinsonisms, including synucleinopathies.” The sense of this sentence is unclear. Do the authors mean “…to discriminate PD from controls and from other causes of parkinsonisms, including synucleinopathies.?

Introduction:

Lines 30-38: “Since the discovery by Polymeropoulos et al. [1], in 1997, of mutations in the gene encoding the presynaptic protein alpha-synuclein (aSyn) as the first mutations related to autosomal dominant Parkinson’s disease (PD), and the description of the presence of aSyn aggregates in Lewy bodies (the pathologic hallmark of PD) [2], which subsequently were also described in other diseases termed “synucleinopathies”, such as PD with dementia (PDD), dementia with Lewy bodies (LBD), and multiple system atrophy (MSA, in this entity, the aggregates were present in the form of glial cytoplasmatic inclusions) [3], many investigators have made important efforts to study the possible role of determinations of this protein in biological fluids and other tissues as a potential biomarker of PD and other 8 synucleinopathies”

This long sentence is hard to read and understand. It should be split into several shorter sentence for easier reading.

Lines 47-48. “The possible value of aSyn determinations in other peripheral tissues in PD, in other synucleinopathies, and other neurological diseases has been the subject (especially in PD) of numerous studies.” The authors should add here a reference on a recent relevant review: ”Biomarkers in Parkinson’s Disease”. Peplow et al., eds. Neurodegenerative Diseases Biomarkers. 2022. Neuromethods, vol 173. pp 155-180. Humana, New York, NY.  https://link.springer.com/protocol/10.1007/978-1-0716-1712-0_7

Lines 112-113:” Several studies have shown similar serum aSyn or antibodies against aSyn levels in patients with Alzheimer’s disease [10, 19, 61, 68, 69, 71] and vascular dementia [68, 71] than those of healthy controls.” Mixing the data on serum alpha-synuclein with the results on antibodies to this protein may lead to confusing results.

Lines 215-216: ”In the same line, Daniele et al. [111] significantly lower aSyn and SNCS/tau heterodimers in 51 patients with AD and 27 with LBD compared with 60 HC.” It looks like that a verb is missed here. Presumably the authors want to say: ”In the same line, Daniele et al. [111] found significantly lower aSyn and SNCS/tau heterodimers in 51 patients with AD and 27 with LBD compared with 60 HC.”

Lines 490-491:”…but a higher and a lower level of two different types of oligomeric aSyn in PD patients. “ The sense of this sentence is unclear. The authors should rewrite it with more details and make it easier to uderatand.

Conclusions and future directions

It would be beneficial if the authors analyze in more details why the data on alpha-synuclein level are often controversial and in what cases the reason of these discrepancies are due to methodological problems.

The authors should avoid very long sentences in order to make the text more reader-friendly.

Author Response

REVIEWER # 2

The following corrections and additions should be made.

Abstract:

Line 24:”…to discriminate PD from controls and form other causes of parkinsonisms, including synucleinopathies.” The sense of this sentence is unclear. Do the authors mean “…to discriminate PD from controls and from other causes of parkinsonisms, including synucleinopathies.?  

OK, this typo was corrected.

Introduction:

Lines 30-38: “Since the discovery by Polymeropoulos et al. [1], in 1997, of mutations in the gene encoding the presynaptic protein alpha-synuclein (aSyn) as the first mutations related to autosomal dominant Parkinson’s disease (PD), and the description of the presence of aSyn aggregates in Lewy bodies (the pathologic hallmark of PD) [2], which subsequently were also described in other diseases termed “synucleinopathies”, such as PD with dementia (PDD), dementia with Lewy bodies (LBD), and multiple system atrophy (MSA, in this entity, the aggregates were present in the form of glial cytoplasmatic inclusions) [3], many investigators have made important efforts to study the possible role of determinations of this protein in biological fluids and other tissues as a potential biomarker of PD and other 8 synucleinopathies” This long sentence is hard to read and understand. It should be split into several shorter sentence for easier reading.

OK, modified to several shorter sentences as follows:

Polymeropoulos et al. [1], in 1997, described mutations in the gene encoding the presynaptic protein alpha-synuclein (aSyn) as the first mutations related to autosomal dominant Parkinson’s disease (PD). In the same year, Spillantini et al. [2] described the presence of aSyn aggregates in Lewy bodies (the pathologic hallmark of PD) [2]. Subsequently, aSyn aggregates were also described in other diseases termed “synucleinopathies”, such as PD with dementia (PDD), dementia with Lewy bodies (LBD), and multiple system atrophy (MSA, in this entity, the aggregates were present in the form of glial cytoplasmatic inclusions) [3]. Since then, many investigators have made important efforts to study the possible role of determinations of this protein in biological fluids and other tissues as a potential biomarker of PD and other synucleinopathies.

Lines 47-48. “The possible value of aSyn determinations in other peripheral tissues in PD, in other synucleinopathies, and other neurological diseases has been the subject (especially in PD) of numerous studies.” The authors should add here a reference on a recent relevant review: ”Biomarkers in Parkinson’s Disease”. Peplow et al., eds. Neurodegenerative Diseases Biomarkers. 2022. Neuromethods, vol 173. pp 155-180. Humana, New York, NY.  https://link.springer.com/protocol/10.1007/978-1-0716-1712-0_7

OK, reference added in the place indicated.

Lines 112-113:” Several studies have shown similar serum aSyn or antibodies against aSyn levels in patients with Alzheimer’s disease [10, 19, 61, 68, 69, 71] and vascular dementia [68, 71] than those of healthy controls.” Mixing the data on serum alpha-synuclein with the results on antibodies to this protein may lead to confusing results.

Ok, we have specified the references regarding total aSyn, oligomeric aSyn, and aSyn antibodies in the text

Lines 215-216: ”In the same line, Daniele et al. [111] significantly lower aSyn and SNCS/tau heterodimers in 51 patients with AD and 27 with LBD compared with 60 HC.” It looks like that a verb is missed here. Presumably the authors want to say: ”In the same line, Daniele et al. [111] found significantly lower aSyn and SNCS/tau heterodimers in 51 patients with AD and 27 with LBD compared with 60 HC.”   

Ok, verb added

Lines 490-491:”…but a higher and a lower level of two different types of oligomeric aSyn in PD patients. “ The sense of this sentence is unclear. The authors should rewrite it with more details and make it easier to uderatand.

OK, corrected.

Conclusions and future directions

It would be beneficial if the authors analyze in more details why the data on alpha-synuclein level are often controversial and in what cases the reason of these discrepancies are due to methodological problems.

Ok, commented in this section

The authors should avoid very long sentences in order to make the text more reader-friendly. Ok, corrected

Reviewer 3 Report

This review is about alpha-synuclein in peripheral tissues as a possible biomarker for neurological diseases.  It is a timely topic, and there is a lot of interest in this topic. However, it has also been covered in a good number of reviews.

The authors do a good job covering studies and the review of the literature has been well updated to the latest published studies. However, I think that the review offers little perspective, but rather a list of research data. I suggest including a discussion section, highlighting the conceptual and technical gaps in the development of alpha-synuclein as a biomarker, with a more extended recommendations to address these challenges as well as the importance of differentiating PD from other synucleinopathies and/or neurological diseases. There are a great variability in the determinations of alpha-synuclein levels in the different tissues and biological fluids and there is not much mentioned or discussed in this term.   

Some other suggestions:

11. In the introduction section, I miss the importance of developing an early biomarker of PD and other synucleinopathies and why there is an increased interest in alpha-synuclein species as potential early biomarkers.      

22. The tittle of the review is “Alpha-synuclein in peripheral tissues as a possible marker for neurological diseases”, so the results of alpha-synucIein determinations in other medical conditions/aging different from neurological diseases are out of scope of this review. Maybe these results could be in a separate section or in the discussion section when you state that subjects with other disease that could influence the results of concentrations of aSyn in peripheral tissues should not be chosen as healthy controls.

33. I suggest following the same structure in all the sections. I mean to structure the sections in terms of diseases (PD, other synucleinopathies and other neurological diseases).

44. There are many studies determining antibodies against alpha-synuclein as well as alpha-synuclein in extracellular vesicles and there is no mention about the relevance of these results in the text.

55. In section 2.3, lines 130-131, there is no data to suggest a role of alpha-synuclein in the development of neuroinflammation and neuronal loss in multiple sclerosis.

Other minor points:

66. There are minor spelling errors throughout the text that Authors should amend before publication (line 24: from instead of form, line 40 / 414 / 420: of instead of or)

77. Line 245: The first description of the presence of Lewy pathology was done by… I suppose that it is the presence of Lewy pathology in salivary glands?

88. Extra spaces between words: lines 98, 121, 160

99. Line 163: use abbreviation for α-synuclein

110.   Line 310: use abbreviation for Alzheimer’s disease

Minor editing of English language required

Author Response

REVIEWER # 3

I suggest including a discussion section, highlighting the conceptual and technical gaps in the development of alpha-synuclein as a biomarker, with a more extended recommendations to address these challenges as well as the importance of differentiating PD from other synucleinopathies and/or neurological diseases. There are a great variability in the determinations of alpha-synuclein levels in the different tissues and biological fluids and there is not much mentioned or discussed in this term.   

Ok, included and commented in the Discussion, conclusions, and future directions section.

Some other suggestions:

  1. In the introduction section, I miss the importance of developing an earlybiomarker of PD and other synucleinopathies and why there is an increased interest in alpha-synuclein species as potential early biomarkers.    

OK, commentary added.  

  1. The tittle of the review is “Alpha-synuclein in peripheral tissues as a possible marker for neurological diseases”, so the results of alpha-synucIein determinations in other medical conditions/aging different from neurological diseases are out of scope of this review. Maybe these results could be in a separate section or in the discussion section when you state that subjects with other disease that could influence the results of concentrations of aSyn in peripheral tissues should not be chosen as healthy controls.

We think that the comments on alpha-synuclein determinations in other medical conditions and aging are interesting and should be included in the text. For this reason, we have changed the title to ALPHA-SYNUCLEIN IN PERIPHERAL TISSUES AS A POSSIBLE MARKER FOR NEUROLOGICAL DISEASES AND OTHER MEDICAL CONDITIONS

  1. I suggest following the same structure in all the sections. I mean to structure the sections in terms of diseases (PD, other synucleinopathies and other neurological diseases).

We have tried to follow a similar structure in all sections, but this is limited by the characteristics of the articles included in each of these sections. We have modified subheadings in sections 3, 5, and 6. In section 2, there are several types of blood cells and those regarding leukocytes and platelets contains a relatively low number of references. Most of the articles included in section 4 cointain a mix of several type of parkinsonisms, and could not be divided following an uniform structure.

  1. There are many studies determining antibodies against alpha-synuclein as well as alpha-synuclein in extracellular vesicles and there is no mention about the relevance of these results in the text.

Ok, we have specified the references regarding total aSyn, oligomeric aSyn, and aSyn antibodies, and mentioned studies determining aSun in extracellular vesicules (exosomes) in the text.

  1. In section 2.3, lines 130-131, there is no data to suggest a role of alpha-synuclein in the development of neuroinflammation and neuronal loss in multiple sclerosis.

We have corrected the sentence to “based on data on previous experimental studies, the authors suggested a possible role of aSyn in the development of neuroinflammation and the diffuse neuronal and synaptic loss [79]”.

Other minor points:

  1. There are minor spelling errors throughout the text that Authors should amend before publication (line 24: frominstead of form, line 40 / 414 / 420: of instead of or)

OK, revised and corrected throughout the text

  1. Line 245: The first description of the presence of Lewy pathology was done by… I suppose that it is the presence of Lewy pathology in salivary glands?

Yes, corrected

  1. Extra spaces between words: lines 98, 121, 160
  2. Revised and corrected throughout the text and tables
  3. Line 163: use abbreviation for α-synuclein

OK, corrected

  1. Line 310: use abbreviation for Alzheimer’s disease

OK, corrected throughout the text.

Round 2

Reviewer 3 Report

The Authors have made an effort to revise all the points. I think that the manuscript is appropiate for publication. The only thing that I still miss is the discussion about the relevance of the determination of alpha-synuclein antibodies as biomarkers instead of different forms of alpha-synuclein as well as the importance of distinguish between PD and other synucleinopathies or neurological disorders. 

Author Response

REVIEWER # 3

  • The only thing that I still miss is the discussion about the relevance of the determination of alpha-synuclein antibodies as biomarkers instead of different forms of alpha-synuclein

OK, We have added the following sentence in the Discussion: “The possible relevance of the determination of serum/plasma aSyn antibodies as reliable biomarkers for PD has been questioned because the great intra-and inter-cohorts variability in the levels of this variable (up to 100-fold variation within groups) and their lack of clear ability to discriminate PD patients from patients with other neurodegenerative diseases [64]”.

  • As well as the importance of distinguish between PD and other synucleinopathies or neurological disorders. 
  1. In our previous version we had mentioned in the introduction the sentence “The development of early biomarkers of PD and other synucleinopathies would be very important to try to establish possible preventive treatments of these diseases”. Now, we have added the following paragraph in the discussion: “Many clinical and experimental data suggest that the pathogenesis processes leading to PD begin several years before the onset of motor symptoms and the clinical diagnosis is made. For this reason, the development of reliable biomarkers (included those related with aSyn) for the early detection of this disease and its differentiation from other synucleinopathies or other neurological diseases, represents an important opportunity to try treatments that could modify the course of the disease [253].”
